# Genetic Algorithm Approach to Design of Multi-Layer Perceptron for Combined Cycle Power Plant Electrical Power Output Estimation

**Ivan Lorencin** 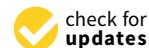**, Nikola Anđelić** , **Vedran Mrzljak \*** and **Zlatan Car**

Faculty of Engineering, University of Rijeka, Vukovarska 58, 51000 Rijeka, Croatia; ilorencin@riteh.hr (I.L.); nandelic@riteh.hr (N.A.); car@riteh.hr (Z.C.)

**\*** Correspondence: vmrzljak@riteh.hr; Tel.: +385-51-651551

**Abstract:** In this paper a genetic algorithm (GA) approach to design of multi-layer perceptron (MLP) for combined cycle power plant power output estimation is presented. Dataset used in this research is a part of publicly available UCI Machine Learning Repository and it consists of 9568 data points (power plant operating regimes) that is divided on training dataset that consists of 7500 data points and testing dataset containing 2068 data points. Presented research was performed with aim of increasing regression performances of MLP in comparison to ones available in the literature by utilizing heuristic algorithm. The GA described in this paper is performed by using mutation and crossover procedures. These procedures are utilized for design of 20 different chromosomes in 50 different generations. MLP configurations that are designed with GA implementation are validated by using Bland - Altman (B-A) analysis. By utilizing GA, MLP with five hidden layers of 80,25,65,75 and 80 nodes, respectively, is designed. For aforementioned MLP, $k$ - fold cross-validation is performed in order to examine its generalization performances. The Root Mean Square Error (*RMSE*) value achieved with aforementioned MLP is 4.305, that is significantly lower in comparison with MLP presented in available literature, but still higher than several complex algorithms such as KStar and tree based algorithms.

**Keywords:** bland-altman analysis; combined cycle power plant; genetic algorithm; machine learning; multi-layer perceptron

## 1. Introduction

A Combined Cycle Power Plant (CCPP) is a power system composed of at least one gas turbine cycle, at least one steam turbine cycle and connection between these cycles – Heat Recovery Steam Generator (HRSG) [1]. Today, engineers and researchers intensively investigate operation and made improvements of such power plants (for CCPPs which are currently in operation), while a new CCPPs are built in many countries worldwide. When CCPPs are compared with other power plants, it can be noticed that the CCPPs are achieving significantly higher efficiencies (usually higher than 60%) [2], with lower specific emmisions [3], and quick start capability (gas turbine cycle) while requiering lower operation and maintenance cost [4]. Due to high complexity of CCPPs, in its investigation and analysis during operation, it is common to use various dynamic numerical simulations for predicting the changes in measured, as well as for calculating non-measured, operating parameters [5].

A literature review offers many analyses of current operating CCPPs. Exergo-economic and environmental analyses of solar integrated CCPP which operates in Poland is presented in [6]. Integration of the solar system into CCPP operation only slightly increases power plant capital cost, but at the same time significantly decreases $CO_2$ emission and therefore $CO_2$ penalties. The investment

return rate is only marginally affected after inclusion of the solar system into CCPP operation. In [7] another exergo-economic analysis of CCPP and investigated three possible scenarios for power plant operation is performed. It is concluded that the optimum size and configuration of the CCPP differ for each observed scenario. Third exergo-economic analysis can be found in [8] where two different CCPP configurations from the same manufacturer were analyzed. Analysis enables selection of better configuration and after selection, the authors performed its optimization and present possibilities of further improvements.

In energy and exergy analyses of CCPPs and its components, several researchers obtained the same conclusions about elements which have the highest losses from both (energy and exergy) aspect. The highest energy losses in CCPP of any kind can be found in steam condenser [9,10], while the highest exergy losses occur in gas turbine combustion chambers [9–11]. These conclusions are confirmed by several researchers and for CCPPs of different size, power output and configuration, so it can be concluded that they are valid in general.

Exergy analysis of any power plant or of any control volume which operate in a power plant is dependable on the ambient pressure and temperature (unlike energy analysis for which the parameters of the ambient are irrelevant) [12,13]. The change in the ambient pressure is usually small and it does not have a major influence on the exergy analysis of power plant or control volume [14], but the change in the ambient temperature can have a major influence on the exergy efficiency and exergy losses of any power plant or control volume. In [15] the ambient temperature change influence on the selected CCPP overall exergy efficiency is investigated. The authors concluded that an increase in the ambient temperature reduces overall CCPP exergy efficiency (increase in the ambient temperature from 8 °C to 23 °C reduces overall exergy efficiency of the analyzed CCPP from 43.3% to 42.7%).

Techniques and recommendations for improving of CCPPs or for improving some components from such power plants can be found in several researches. In [16] an analysis of the modern CCPP in which gas turbine uses steam cooling is presented. Proposed cooling technique increases CCPP overall efficiency, while simultaneously, such technique reduces plant flexibility and increases power plant start-up time. In [17] various gas turbine improvements in a modern CCPP are analyzed and is concluded that industry - known solutions such as sequential combustion can significantly increase overall plant efficiency.

The authors in [18,19] investigate the benefits of steam injection into the combustion chambers of gas turbine which is a constituent component of CCPP. The main conclusions are that such improvement increases power output of CCPP, decreases plant $NO_x$ emission (due to decreasing of the maximum combustion temperature) and provide acceptable economic performance.

In [20] a new operating strategy for improving part-load performance of analyzed CCPP is presented. Proposed strategy resulted with an increase in CCPP overall efficiency up to 1.2% at partial loads. The possibility of a wind farm integration into an offshore CCPP is investigated in [21]. The authors found many difficulties in such integration because many, possibly conflicting requirements have to be satisfied simultaneously.

Analysis of the water amount reduction in CCPP cooling systems was performed in [22]. Three different cooling systems were analyzed—wet, dry, and hybrid (the wet system uses water, and the dry system uses air circulated by a fan, while the hybrid system is an alternative which combines wet and dry techniques). The hybrid cooling system has the highest investment costs, but it also provides many benefits in comparison to other observed cooling systems.

In recent research papers, it can be noticed that the authors prefer two improvements of CCPPs which today bring the largest benefits into such power plants operation. The first is integration of solar systems and the second is integration of $CO_2$ post-combustion capture systems into CCPP operation. A computational analysis of small solar field integration in CCPP operation is presented in [23]. When compared to base CCPP, small solar field integration increases plant overall efficiency for about 2.58% in the morning and afternoon periods and for about 3.16% in the midday periods. Another mathematical model of a typical solar integrated CCPP is developed in [24] and applied on Al

- Abdaliya's solar integrated CCPP in Kuwait. Mathematical model results show that the observed solar integrated CCPP can reach an overall efficiency of more than 66%.

A post-combustion $CO_2$ capture system which use activated carbon and its comparison with commercial systems applied in CCPPs operation is investigated in [25]. After performing comparisons, it was concluded that a system which uses activated carbon can be a good alternative for $CO_2$ capture and such a system can be more efficient and cost beneficial in comparison with other commercial systems. Another alternative to commercial systems for $CO_2$ capture in CCPPs, named the Moving Bed Temperature Swing Adsorption (MBTSA) system, was presented and analyzed in [26]. As well as for activated carbon system, for MBTSA system is also concluded to represent a newer, more proper alternative for $CO_2$ capture in CCPPs, and it brings several advantages in comparison with other $CO_2$ capture systems.

Post-combustion $CO_2$ capture system along with methanation system and its implementation into CCPP operation in India was analyzed in [27]. Captured $CO_2$ in this combination is used in methanation system to produce methane – produced methane is used as a fuel in a gas turbine. In comparison with base CCPP, implementation of these two systems resulted with a significant increase in plant power output.

Unlike other research papers which investigated CCPPs and its components, in [28] a comparison analysis of different machine learning (ML) techniques for prediction of CCPP full load electrical power output is presented. This article, as well as the resulting dataset, was used as a starting point for research performed in this paper. The dataset presented in [28] was published online as part of the UCI Machine Learning Repository. Overview of the methods presented in aforementioned article and achieved *RMSE* is given in Table 1.

**Table 1.** Overview of methods and achieved *RMSE*.

| Category | Method | *RMSE* |
| --- | --- | --- |
| Functions | Simple Linear regression | 5.425 |
| | Linear Regression | 4.561 |
| | Least Median Square | 4.968 |
| | Multilayer Perceptron | 5.341 |
| | Radial Basis Funcion Neural Network | 7.501 |
| | Pace Regression | 4.561 |
| | Support Vector Poly Kernel Regression | 4.563 |

When presented results are compared, it can be observed that Artificial Neural Networks (ANNs) have significantly higher *RMSE* compared to other methods, even when compared to simple regression functions. This feature is also noticeable when using an MLP.

In the energy sector, Artificial Neural Networks (ANNs) are widely applied for resolving many problems and for optimization purposes. In [29] ANN-based reinforcement learning algorithm is used to manage the optimal energy routing path in energy internet (EI) concept. In order to effectively analyze the quality of power signals, research [30] proposes a method of signal feature capture and fault identification based on the ANN combined with discrete wavelet transform and Parseval's theorem. ANN for air-temperature predictions in smart buildings was developed in [31] in order to obtain better energy control. Short term forecasting prediction of the photovoltaic plant power output by using ANNs can be found in [32,33]. Analysis of heating expenses in a large social housing stock using artificial neural networks is presented in [34]. Energy supply solution for sensor nodes in buildings based on ANNs was investigated and analyzed in [35]. ANNs can be also used for fast and precise detection of garbage patches, oil spills or pollutions of any kind inside each energy plant or in the entire geographical area with several energy plants by using aerial imagery [36].

It is shown that ANNs are, in general, performing better in comparison with other regression methods [37] and it can be considered as a more flexible method [38] that is acceptable from time standpoint. MLP can be considered as the most used type of ANN [39] due to high performance from the standpoint of regression [40,41] and classification [42,43]. Because of these reasons, the aim of this research is to find a suitable solution to MLP model selection problem, that is applicable to prediction of CCPP electrical power output. Furthermore, the aim of this research is to find MLP model that performs with lower *RMSE*, in comparison to ANNs presented in [28].

There are many approaches to solving ANN model selection problem such as: grid search [44,45], Bayesian model selection [46], etc. Another method for solving MLP model selection problem is a heuristic approach [47–49]. This approach offers high performances in regard of regression [50,51] and classification [52] problems. For these reasons a utilization possibility of heuristic algorithms in design of MLP for CCPP electrical power output estimation is investigated.

During analysis and optimization of energy systems, various heuristic algorithms can be used. However, the guidelines which will lead researchers to selection of optimal optimization algorithms for investigated (or similar) problem cannot be found in the literature. Therefore, the best selection procedure is to use an optimization algorithm which is used by other researchers during investigation of similar problems, or during analysis of similar systems. In this paper, the authors selected Genetic Algorithm (GA) for optimization of MLP neural network architecture for Combined Cycle Power Plant (CCPP) electrical power output estimation. Similar research of energy management optimization at building and district levels is performed in [53] where the authors used ANN and GA. However, the authors in [53] used GA for optimization of ANN predictions, not for optimization of ANN architecture. Literature review offers many examples of using GA in analysis and optimization of various elements from many energy systems or its parts. In [54] as well as in [55] GA is utilized for optimization of energy systems which uses solar energy sources. Optimal energy management of a stand-alone hybrid energy system by using GA strategies is presented in [56], while energy quality management for a micro-energy network integrated with renewables in a tourist area was analyzed in [57] where the authors used GA optimization in order to obtain optimal energy distribution. Reducing of water pumps electricity usage and pollution emissions by using sorting GA can be found in [58]. From presented literature, it can be concluded that the various researchers often used GA in investigation, analysis and optimization of energy systems.

In the analysis and optimization of energy systems or its parts besides GA, other optimization algorithms can be utilized. Gravitational Search Algorithm (GSA) is one of such optimization algorithms used in optimization of pumped storage hydro unit [59], in the forecasting of coal demand [60] or in forecasting of monthly electricity demands [48] as well as in other energy and engineering problems [61]. Particle Swarm Optimization (PSO) algorithm is used so far for developing of power loss reduction method in power industry sector [62], in optimization of power supply system [63] as well as in the analysis and optimization of smart power grids [64]. Ant Colony Optimization (ACO) can be used in optimization of biodiesel production [65], in AC/DC distribution network planning problem [66], etc. Cuckoo Search Algorithm (CSA) usage is found in the analysis and optimization of magnetic levitation system [67], in route optimization of heating engineering [68] as well as in research and application of hybrid wind-energy forecasting models [69]. Hybrid Genetic Algorithm (HGA) is improved version of classic GA, which also can be used in resolving many problems of energy systems. For example, HGA can be used in developing of control schemes for small power systems with high-penetration wind farms [70] or in the thermal fatigue failure prediction of microelectronic chips [71]. In addition to the aforementioned research papers, in the literature, several other optimization algorithms used in energy and other practical applications can be found [72,73].

The direct performance comparison of several optimization algorithms can be found in a few research papers. In [74] were compared performances of several optimization algorithms (PSO, ACO, CSA, GA and HGA) while performing optimization of real-time task scheduling in multiprocessor systems. For this problem the authors concluded that the best performance shows HGA, following by

GA and other algorithms. In [75] the authors compared four algorithms (Simulated Annealing, GA, HGA and Variable Search Environment Descending) while solving problem of electric power grids distribution for optimal location and sizing. It is concluded that HGA (followed by GA) provides solutions of the best quality, with a note that HGA uses significantly higher computational time in comparison with other methods. Therefore, for solving the same problem, it is impossible to conclude which algorithm is absolutely dominant. A performance comparison of several optimization algorithms while solving residential load scheduling problem is presented in [76]. This research also confirmed that GA in comparison to all other algorithms have advantages and disadvantages, but that it always give satisfying results along with using a reasonable amount of computational resources.

Presented literature review shows that GA can be used for optimization of many elements and problems in various energy systems, therefore it is also chosen for the research performed in this paper as a reliable and fast optimization solution for which can be expected to give satisfactorily accurate and precise results.

In this research, a GA approach to design of MLP for CCPP electrical power output estimation is presented. GA - based MLP model selection is performed for MLP with one, two, three, four and five hidden layers using one mutation procedure, three different crossover procedures and two different fitness functions. For obtained results, Bland-Altman (B-A) analysis is performed and three MLP configurations are chosen. Regression results achieved with aforementioned MLP configurations are than compared to real data. At the end, *RMSE* values of MLPs designed with GA are compared to results presented in [28].

To summarize the novelty of this paper, the idea is to investigate the implementation possibility of heuristic algorithms, mainly GA, in order to increase regression performances of MLP for CCPP electrical power output estimation, in comparison to results presented in Table 1. As an addition to classification performance measures, B-A analysis is introduced alongside *RMSE* in order to examine standard derivation of errors produced by regression. For chosen configurations, cross-validation is performed in order to investigate generalization performances of aforementioned configurations. From previous statements, the following hypotheses can be imposed:

- to investigate implementation possibility of GA in design of MLP for CCPP electrical power output estimation,
- to compare regression performances of GA - designed MLP with results presented in available literature and
- to determine MLP configuration with optimal performances in regard of regression errors.

Based on presented hypotheses, optimal MLP configuration will be presented and possibility of heuristic algorithms utilization in design of MLP for CCPP electrical power output estimation will be discussed.

## 2. Materials and Methods

In this section an overview of used materials and methods is provided. First, a brief description of dataset is presented. After this part, description of used MLP and GA is given. At the end, methods used for results comparison are described.

### 2.1. Dataset Description

Analyzed combined cycle power plant has two identical gas turbines with the same operating parameters and consequentially the same produced power. Steam turbine is composed of high-pressure and low-pressure cylinders mounted on the same shaft, with a note that the low-pressure cylinder is a dual flow symmetrical cylinder, as shown in Figure 1.

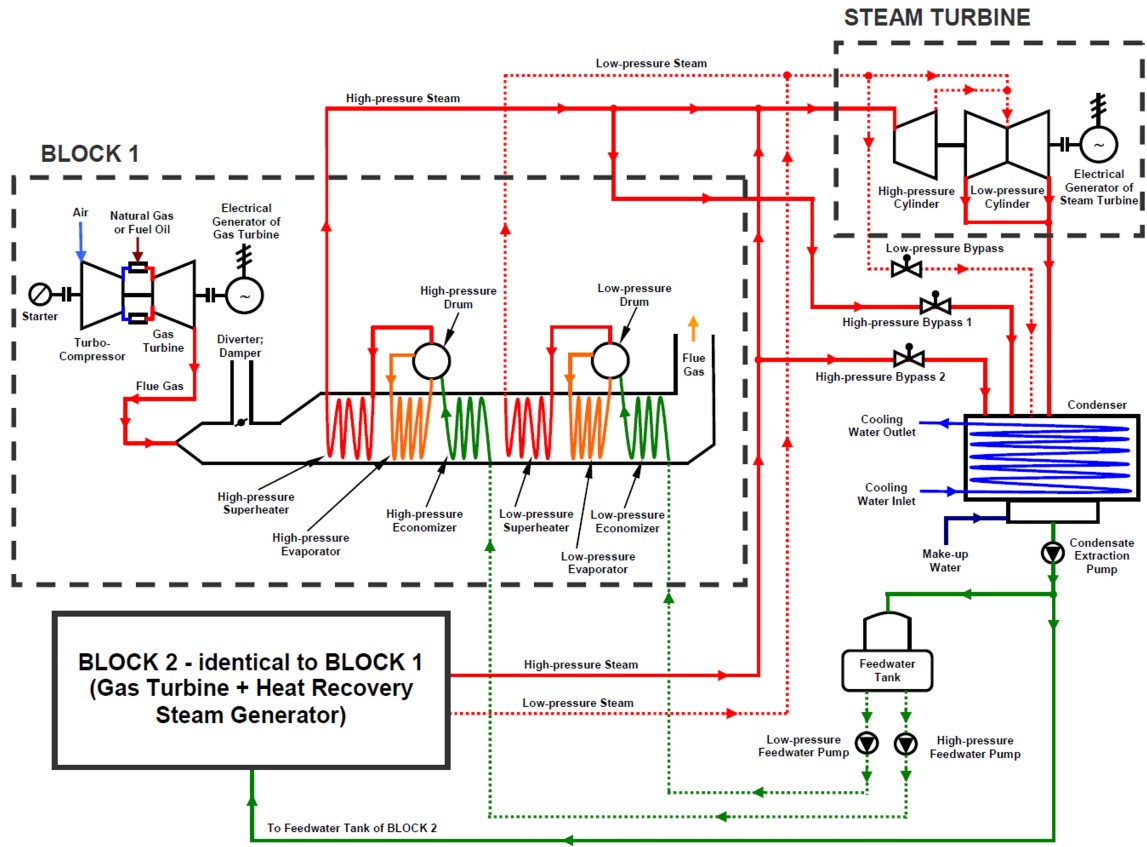

**Figure 1.** Schematic diagram of the analyzed CCPP.

Cumulative power produced from the analyzed combined cycle power plant in each operating regime can be calculated by using two different approaches: the first is conventional approach, while the second is approach by applying Machine Learning (ML) algorithms in order to estimate electrical power output. The conventional approach of cumulative produced power calculation from the analyzed plant in each operating regime requires knowledge of 20 operating parameters (operating medium pressure, temperature and mass flow rate) in each characteristic operating point of gas turbine and steam turbine. Such number of operating parameters in a conventional approach requires extensive measurements in each plant operating regime. The second approach, by using ML algorithms, before its final implementation, requires knowledge of only five operating parameters in each plant operating regime. Those parameters are ambient air pressure, temperature and relative humidity; condenser pressure (vacuum); and cumulative electrical power output of the CCPP. After implementation ML - based algorithms require knowledge of only four operating parameters (ambient air pressure; temperature; relative humidity and condenser vacuum) for the calculation of cumulative electrical power output of the CCPP in each operating regime. Parameters used for training and testing of ML algorithms are found in [28] and described in Table 2.

Dataset used for this research consists of 9568 data points (CCPP operating regimes) that are constructed with four-element input vector and output value. In this research, dataset is divided in two subsets: training and testing dataset. Training dataset consists of 7500 data points and testing dataset consists of 2068 data points. Training dataset will be used for MLP training, while testing dataset will be used for MLP testing and fitness value determination. Such dataset division is also called fixed partitions division [77]. Furthermore, testing dataset will be used for Bland-Altman analysis and for *RMSE* calculation with aim for determination of configurations that are achieving viable estimation results. For aforementioned configurations cross-validation will be performed with aim to examine stability in regard of data generalization. In this research, cross-validation will be performed on already designed configurations and not during GA - based MLP design procedure. The reason for this lies

in the fact that performing cross-validation during GA will be time consuming due to dataset and population sizes [78].

**Table 2.** Description of dataset parameters.

| Type | Parameter | Range |
|------|-----------|-------|
| Input | Temperature ($T$) | 1.81 °C–37.11 °C |
| Input | Ambient Pressure ($AP$) | 992.89 mbar–1033.30 mbar |
| Input | Relative Humidity ($RH$) | 25.56%–100.16% |
| Input | Exhaust Vacuum ($V$) | 25.36 cmHg–81.56 cmHg |
| Output | Average Hourly Electrical Power Output ($P_e$) | 420.26 MW–495.76 MW |

As mentioned before, a $k$ - fold cross-validation technique is utilized in order to examine generalization performances of designed MLPs. First step for performing $k$ - fold cross-validation is division of the entire dataset into $k$ parts [77,79]. Than, one part of the divided dataset is used for MLP testing, while remaining parts are used for MLP training. Graphical representation of such procedure is shown in Figure 2.

**Figure 2.** Graphical representation of the cross-validation procedure ($F_1$ - Fold 1; $F_2$ - Fold 2; $F_3$ - Fold 3; $F_4$ - Fold 4; $F_5$ - Fold 5).

In this research, two types of $k$ - fold cross-validation are performed and these are 5-fold and 10-fold cross-validation.

*2.2. Multilayer Perceptron*

MLP is a type of ANN, characterized with feed-forward architecture and according to [80] mostly consists of three layers and these are:

- Input layer - layer which represents input data vector,
- Hidden layers - layers between input and output layer and
- Output layer - layer that represents output vector.

This MLP architecture is characterized with its simple design, that can be used for solving various classification and regression problems [80]. Each hidden layer consists of nodes constructed with some type of activation function, that is used for transforming summed input value of each neuron to its output value [81]. For the case of this research, three different activation functions are utilized, and these are:

- Rectified Linear Unit (ReLU) [82],
- Logistic Sigmoid [83] and
- Hyperbolic Tangent (Tanh) [84].

In order of defining relation between data in input vector and data in output vector, procedure called supervised learning is performed [85]. Supervised learning is achieved by using an algorithm for optimization (solver). In the case of this research, six different solvers are utilized, and these are:

- Stochastic Gradient Descent (*SGD*) [86],
- Adaptive learning rate optimization algorithm (*Adam*) [87],
- Root-Mean-Square optimization algorithm (*RMSProp*) [88],
- Adaptive Gradient Algorithm (*AdaGrad*) [89] ,
- Proximal AdaGrad (*PAdaGrad*) [90] and
- Ada Delta (*Ada*Δ) [91].

By using above mentioned activation functions and solvers GA-based procedure of MLP design will be performed.

### 2.3. Genetic Algorithm

GA is a meta-heuristic algorithm used for optimization [92], path planing [93] and mapping [94] tasks. It is inspired by the process of natural evolution and it is based on procedures similar to ones in natural evolution process used for generating individual solutions with better performances [95]. These procedures are:

- Mutation,
- Crossover and
- Parents selection.

Procedures listed above will be used on chromosomes that represent MLP parameters. The following part of the paper represents a description of the previously mentioned methods of variation and selection, that are used in this research. A definition of fitness function and description of population initialization are also provided.

#### 2.3.1. Mutation

Number of mutations ($N$) into one chromosome is a uniformly distributed discrete random variable, which can be described as

$$N \sim U[0, M], \tag{1}$$

where $M$ represents chromosome length. Location of the mutated gene into the chromosome ($G$) is also defined as a uniformly distributed ($U$) random variable

$$G \sim U[0, M]. \tag{2}$$

The new parameter ($a$) contained in the gene ($G$) is selected randomly as

$$S_G \xleftarrow{\text{R}} a, \tag{3}$$

where $S_G$ represents a set of parameters that are applicable to gene $G$. Random selection of a set $S_G$ member which is assigned to the gene $G$ is also uniformly distributed. A graphical representation of mutation procedure with $M = 10$ and $N = 3$ is given in Figure 3, where shaded cells in $K_3$ represent mutated genes.

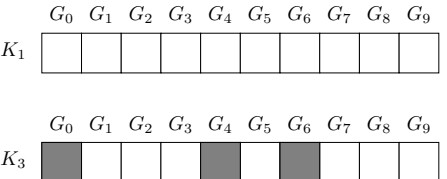

**Figure 3.** Graphical representation of the mutation procedure on the example with $M = 10$ and $N = 3$ ($K_1$ - parent chromosome, $K_3$ - child chromosome, $G$ - gene).

2.3.2. Crossover

In this research, three different crossover methods are utilized. Determination of a crossover method is performed by using uniformly distributed random variable, defined as

$$c \sim U[0, 2], \tag{4}$$

where $c$ represents a random variable. Depending on the value of the random variable, the crossover method is defined, as shown by the random selection tree in the Figure 4a.

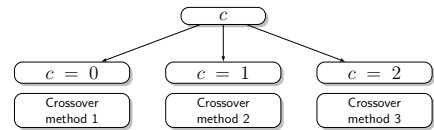

(a) The random selection tree of the crossover method

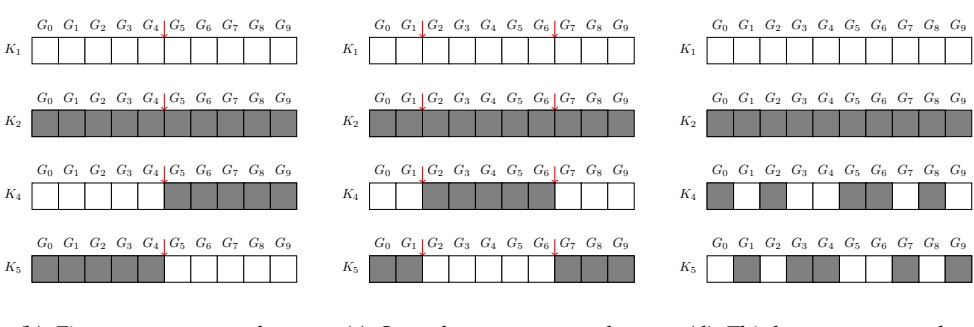

(b) First crossover procedure  (c) Second crossover procedure  (d) Third crossover procedure

**Figure 4.** Graphical representation of used random selection tree (**a**) and used crossover methods (**b–d**) ($K_1$ - first parent chromosome, $K_2$ - second parent chromosome, $K_4$ - first child chromosome, $K_5$ - second child chromosome, $G$ - gene).

The first crossover method is performed with one crossover of two different chromosomes. Gene after which a crossover is performed is defined as a uniformly distributed discrete random variable which can be written as:

$$G \sim U[0, M - 1]. \tag{5}$$

The fragment of each of the parent chromosomes is chosen to form two new chromosomes. First child chromosome ($K_4$) is formed by combining the first fragment of the first parent chromosome ($K_1$) and the second fragment of the second parent chromosome ($K_2$). Formation of the first child chromosome can be defined as

$$K_4 = K_1\{0, 1, ..., G\} \cup K_2\{G + 1, G + 2, ..., M\}. \tag{6}$$

On the other hand, second child chromosome is formed by combining the first fragment of the second parent chromosome ($K_2$) and the second fragment of the first parent chromosome ($K_1$). This formation can be represented with

$$K_5 = K_2\{0, 1, ..., G\} \cup K_1\{G + 1, G + 2, ..., M\}. \tag{7}$$

Graphical representation of previously defined crossover procedure is shown in Figure 4b. The second crossover method utilized in this research is performed by combining three different fragments of two parent chromosomes for construction of two child chromosomes. Parent chromosomes are divided into three fragments by using two discrete, uniformly distributed, random variables representing genes after which chromosome division has occurred. First variable represents gene that forms the end of the first fragment of the chromosome. That chromosome ($G_l$), which can be considered as a left borderline chromosome, is defined as

$$G_l \sim U[0, M - 2], \tag{8}$$

where $M - 2$ marks the farthest possible boundary, so defined, that by the end of the child chromosome, two more fragments of the first and second parent chromosomes could be exchanged. Second, the right chromosome $G_r$, is defined as

$$G_r \sim U[G_l + 1, M - 1], \tag{9}$$

where at least one gene is left to be fitted with a genetic material of the first parent chromosome. The first child chromosome ($K_4$) is constructed by combining first fragment of the first parent chromosome ($K_1$), middle fragment of the second parent chromosome ($K_2$) and the last fragment of the first parent chromosome ($K_1$). This combination can be written as

$$K_4 = K_1\{G_0, G_1, ..., G_l\} \cup K_2\{G_l + 1, G_l + 2, ..., G_r\} \\ \cup \quad K_1\{G_r + 1, G_r + 2, ..., M\}. \tag{10}$$

Formation of the second child chromosome ($K_5$) is performed by using opposite chromosome fragments. $K_5$ is formed by combining the first fragment of the second parent chromosome ($K_2$), the middle fragment of the first parent chromosome ($K_1$) and last fragment of the second parent chromosome. Formation of the second child chromosome can be written as

$$K_5 = K_2\{G_0, G_1, ..., G_l\} \cup K_1\{G_l + 1, G_l + 2, ..., G_r\} \\ \cup \quad K_2\{G_r + 1, G_r + 2, ..., M\}. \tag{11}$$

The graphical representation of the second crossover procedure is given in Figure 4c. Third crossover procedure can be defined with discrete, uniformly distributed random variable ($b$) that, in fact, represents a coin-flip random variable, which can be written as

$$b = U \sim [0, 1]. \tag{12}$$

According to the value of variable $b$, two child chromosomes are constructed. A gene of the first child chromosome ($K_{4i}$) will be equal to the gene of the first parent chromosome ($K_{1i}$) if $b = 0$. On the other hand, $K_{4i}$ will be equal to the gene of the second parent chromosome ($K_{2i}$) if $b = 1$. The above procedure can be defined with

$$K_{4i} = \begin{cases} K_{1i}, & b = 0 \\ K_{2i}, & b = 1 \end{cases}, \tag{13}$$

for $i = 0, 1, ..., M$. Procedure for constructing the second child chromosome is performed in the similar manner. A gene of the second child chromosome ($K_{5i}$) will be equal to the gene of the second parent chromosome ($K_{2i}$) if $b = 0$. On the other hand, $K_{5i}$ will be equal to the gene of the first parent chromosome ($K_{1i}$) if $b = 1$. As it is in the case of constructing $K_{4i}$, construction of $K_{5i}$ can be defined with

$$K_{5i} = \begin{cases} K_{2i}, & b = 0 \\ K_{1i}, & b = 1 \end{cases}, \tag{14}$$

for $i = 0, 1, ..., M$. The graphical representation of the described crossover procedure is given in Figure 4d.

### 2.3.3. Fitness Function

Fitness function is a function that is used as a chromosome performance measure and it is used to guide the GA towards the optimal solution [96]. In this research, two different fitness functions are defined, and these are

- $\overline{MRE}$ and
- $\overline{MSE}$.

As mentioned previously, fitness function is a function that represents a chromosome performance measure. From this statement it can be concluded that fitness value is a function of a chromosome, defined as

$$F_i = f(K_i),\tag{15}$$

where $K_i$ represents a chromosome. One way to define a fitness value of a chromosome is to calculate $\overline{MRE}$ of a sample $e_i$ which can be defined with

$$e_i = \frac{|Y_i - \hat{Y}_i|}{Y_i},\tag{16}$$

where $Y_i$ represents a real value of an sample and $\hat{Y}_i$ represents a value that is predicted with an MLP-based regression. $\overline{MRE}$ of the entire testing dataset can be calculated with

$$\overline{MRE} = \frac{1}{N}\sum_{i=1}^{N}\frac{|Y_i - \hat{Y}_i|}{Y_i},\tag{17}$$

where $N$ represents the total number of samples in the testing dataset.

As an another approach of determining fitness value of a chromosome, square error is defined and on a sample ($SE_i$) it can be calculated as

$$SE_i = (Y_i - \hat{Y}_i)^2,\tag{18}$$

where $Y_i$ represents real output value and $\hat{Y}_i$ represents a value predicted with MLP-based regression. Mean square error of the entire testing dataset ($\overline{MSE}$) can be calculated as

$$\overline{MSE} = \frac{1}{N}\sum_{i=1}^{N}(Y_i - \hat{Y}_i)^2,\tag{19}$$

where $N$ represents the total number of samples of composed testing dataset.

Both methods described above will be used to determine the fitness value of chromosomes.

### 2.3.4. Population Creation

Initial population $P$ is created by combining chromosomes with randomly chosen genes, which can be written as

$$P = \{K_I, K_{II}, ..., K_\rho\},\tag{20}$$

where $\rho$ represents the number of population members. For each population member a fitness value is determined by using one of methods described in Section 2.3.3. Fitness value of each chromosome is than used to form a set that contains fitness values of all population members. This set can be defined as

$$F = \{F_I, F_{II}, ..., F_\rho\}.\tag{21}$$

By using set of chromosomes $P$ and set of fitness values $F$, a new set of tuples is constructed. This set can be defined with

$$S = \{(K_I, F_I), (K_{II}, F_{II}), ..., (K_\rho, F_\rho)\},\tag{22}$$

where each of the tuples represents one chromosome and its fitness value.

### 2.3.5. Parents Selection

In order to perform selection of parents for formation of new generation,previously defined set of tuples can be written in following form

$$S = \{S_I, S_{II}, ..., S_\rho\}, \tag{23}$$

is sorted in such manner that

$$\pi_2(S_I) \leq \pi_2(S_{II}) \leq ... \leq \pi_2(S_\rho), \tag{24}$$

where $\pi_2(S_i)$ represents the second element of a tuple $S_i$, namely $F_i$. By using the sorted set, parent chromosomes are determined as

$$K_1 = \pi_1(S_I) \tag{25}$$

and

$$K_2 = \pi_1(S_{II}), \tag{26}$$

where $\pi_1(S_i)$ represents the first element of the tuple $S_i$, namely $K_i$.

### 2.3.6. New Population Formation

After selection of parents ($K_1$ and $K_2$), variation procedures are performed. Parents together with children generated by variation procedures ($K_3$, $K_4$ and $K_5$) form the new population, that can be defined as a union between four sets

$$P = \begin{matrix} \{K_1, K_2\} \cup \{K_{31}, K_{32}, ..., K_{3D}\} \cup \{K_{41}, K_{42}, ..., K_{4C}\} \\ \cup \quad \{K_{51}, K_{52}, ..., K_{5C}\}, \end{matrix} \tag{27}$$

where $D$ represents number of chromosomes produced with mutation and $C$ represents half of the number of performed crossover procedures. From population set, number of population members can be defined as

$$\rho = 2 + D + 2C. \tag{28}$$

In the case of this research GA parameters $C = 6$ and $D = 6$ are utilized, that gives 20 population members in total. The described procedure is performed for 50 generations.

### 2.3.7. Chromosome Construction

For the purposes of this research six different MLP parameters are utilized in chromosome construction. Parameters that can be described as numerical values are shown in Table 3. As a number of MLP training epochs, an odd number between one and 100 is chosen. As a number of nodes in each hidden layer an integer in range from 10 up to 100 is chosen. These constrains were introduced with regard to theoretical knowledge of model complexity selection [97,98]. As a batch size, a value in range from 200 up to 1875 samples is chosen, where the latter represents a quarter of a number of training samples.

**Table 3.** Numerical parameters used for chromosome construction.

| Parameter | Variations | Range |
|---|---|---|
| Number of epochs ($l_e$) | 50 | $l_e \in [1, 100]$ |
| Number of nodes in the hidden layer ($l_n$) | 20 | $l_n \in [10, 100]$ |
| Batch size ($l_b$) | 75 | $l_b \in [200, 1875]$ |

Other two parameters used in chromosome construction are represented with sets of strings, as shown in Table 4. The first set represents a set of activation functions used for MLP design where activation functions are used for hidden layers and output layer design, while the second set consists of different solver algorithms that are used for MLP training.

**Table 4.** String parameters used for chromosome construction.

| Parameter | Set |
|---|---|
| Activation function ($l_a$) | $l_a = \{ReLU, Logistic\ Sigmoid, Tanh\}$ |
| Solver ($l_s$) | $l_s = \{SGD, Adam, RMSprop, AdaGrad, PAdaGrad, Ada\Delta\}$ |

Parameters used for construction of chromosome in the case of MLP with one hidden layer are shown in Table 5. It can be noticed that set $l_a$ is used for the generation of two genes $G_2$ and $G_3$ were $G_2$ represent activation function used for the design of the hidden layer and gene $G_3$ represents the activation function used for the design of the output layer.

**Table 5.** Chromosome construction parameters for the case od MLP with one hidden layer.

| Gene | Gene Representation | Parameter |
|---|---|---|
| $G_0$ | Number of epochs | $l_e$ |
| $G_1$ | Number of neurons in the hidden layer | $l_n$ |
| $G_2$ | Activation function in the hidden layer | $l_a$ |
| $G_3$ | Activation function in the output layer | $l_a$ |
| $G_4$ | Batch size | $l_b$ |
| $G_5$ | Solver | $l_s$ |

The chromosome construction parameters used in the case of MLP designed with two hidden layers are shown in Table 6. It can be seen that each of the hidden layers is defined with its own activation function and its own number of nodes.

**Table 6.** Chromosome construction parameters for the case od MLP with two hidden layers.

| Gene | Gene Representation | Parameter |
|---|---|---|
| $G_0$ | Number of epochs | $l_e$ |
| $G_1$ | Number of neurons in the first hidden layer | $l_n$ |
| $G_2$ | Activation function in the first hidden layer | $l_a$ |
| $G_3$ | Number of neurons in the second hidden layer | $l_n$ |
| $G_4$ | Activation function in the second hidden layer | $l_a$ |
| $G_5$ | Activation function in the output layer | $l_a$ |
| $G_6$ | Batch size | $l_b$ |
| $G_7$ | Solver | $l_s$ |

The same pattern is repeated for the cases of MLP designed with three, four and five hidden layers.

*2.4. Bland-Altman Analysis*

For regression model validation, B-A analysis is used. B-A plot is a statistical tool mostly used for comparison of medical measurement methods [99,100], but it can also be used to validate machine learning-based regression [101,102]. For every sample contained in testing dataset, a point on B-A plot can be determined as

$$T(p_i, d_i),\tag{29}$$

where $p_i$ can be defined as a mean value between real and estimated CCPP electrical power output

$$p_i = \frac{Y_i + \hat{Y}_i}{2}\tag{30}$$

and $d_i$ can be defined as a difference between two samples (real and estimated)

$$d_i = Y_i - \hat{Y}_i.\tag{31}$$

Using described procedure, a graphical representation of all samples in testing dataset is performed. By using aforementioned samples, method comparison metrics are introduced, and these are:

- Bias,
- Confidence interval upper bound and
- Confidence interval lower bound.

The mean value of differences or bias can be determined with

$$d = \frac{1}{N}\sum_{i=1}^{N} d_i,\tag{32}$$

where $N$ represents the number of samples in testing dataset. By using calculated bias, lower and upper bounds of the confidence interval are determined as

$$LOA_l = d - 1.96s_d\tag{33}$$

and

$$LOA_u = d + 1.96s_d,\tag{34}$$

where $s_d$ represents standard deviation, calculated as

$$s_d = \sqrt{\frac{1}{N-1}\sum_{i=1}^{N}(d_i - d)^2}.\tag{35}$$

The value 1.96 corresponds to 95% confidence interval. By using aforementioned metrics, MLP that has the best match to the samples from the testing dataset will be determined.

*2.5. Root Mean Square Error*

For MLP comparison with results presented in [28], *RMSE* values are used. The *RMSE* of a testing dataset is calculated as:

$$RMSE = \sqrt{\frac{1}{N}\sum_{i=1}^{N}(Y_i - \hat{Y}_i)^2}.\tag{36}$$

## 3. Results and Discussion

In this section MLP configurations that are results of GA - based MLP design are presented, along with their fitness value. All presented configurations are designed by utilizing above described GA procedure in 50 generations. For all resulting configurations, B-A statistical analysis is performed with aim of comparing estimated results with real data. Results of B-A analysis will be used for evaluation of regression performances of each MLP configuration designed by utilizing GA. According to results achieved with B-A statistical analysis, some of the configurations will be chosen for further analysis, while the others will be omitted. Performances of remaining configurations will be compared to results achieved by similar methods that are presented in [28]. In the end, cross-validation technique will be applied to the remaining configurations with the aim of determining the generalization performances of each. According to obtained results, MLP configuration with optimal performances will be determined.



### 3.1. Results

When GA with $\overline{MRE}$ fitness function is utilized for MLP design, configurations reported in Table 7 are obtained.

**Table 7.** MLP configurations as a result of the GA implementation with $\overline{MRE}$ as a fitness function.

| | Configuration | | | | |
|---|---|---|---|---|---|
| **Gene** | $R_1$ | $R_2$ | $R_3$ | $R_4$ | $R_5$ |
| Number of epochs | 87 | 70 | 99 | 91 | 13 |
| Number of neurons in the first hidden layer | 60 | 80 | 65 | 55 | 80 |
| Activation function in the first hidden layer | ReLU | ReLU | Sigmoid | ReLU | Sigmoid |
| Number of neurons in the second hidden layer | - | 100 | 10 | 55 | 25 |
| Activation function in the second hidden layer | - | ReLU | ReLU | Sigmoid | Tanh |
| Number of neurons in the third hidden layer | - | - | 60 | 90 | 65 |
| Activation function in the third hidden layer | - | - | ReLU | Sigmoid | ReLU |
| Number of neurons in the fourth hidden layer | - | - | - | 85 | 75 |
| Activation function in the fourth hidden layer | - | - | - | ReLU | ReLU |
| Number of neurons in the fifth hidden layer | - | - | - | - | 80 |
| Activation function in the fifth hidden layer | - | - | - | - | ReLU |
| Activation function in the output layer | ReLU | ReLU | ReLU | ReLU | ReLU |
| Batch size | 407 | 1164 | 453 | 223 | 39 |
| Solver | Adam | Adam | Adam | SGD | Adam |
| Fitness value ($\overline{MRE}$) | 0.8327% | 1.0062% | 0.7374% | 1.4606% | 0.7369% |

From results presented in Table 7 it can be noticed that MLP designed with one hidden layer ($R_1$) achieves fitness value of 0.8327%. When B-A analysis is performed on results achieved with $R_1$ it can be seen that this MLP is performing with negative bias of $-2.6766$ and confidence interval in range $d \in [-12.4106, 7.0574]$, as shown in Figure 5a. It can be noticed that points on B-A plot are concentrated around bias line, regardless of average value.

In the case of GA usage for design of MLP with two hidden layers ($R_2$), the configuration reported in Table 7 is obtained. This configuration achieves fitness value of 1.0062%. When Bland-Altman analysis is performed for $R_2$, it can be noticed that MLP, in this case, is performing with positive bias of 14.7462 and confidence interval in range $d \in [3.5776, 25.9149]$. It can also be noticed that MLP is performing with significantly higher error, in comparison to the MLP designed with one hidden layer. This feature is particularly emphasized in the case of samples with higher mean value, as shown in Figure 5b.

In the case of MLP with three hidden layers ($R_3$), configuration reported in Table 7 is obtained. With this configuration, fitness value of 0.7374% is achieved. When B-A analysis is performed for $R_3$, it can be seen that all errors are linearly distributed with a positive slope, relative to the mean values, as shown in Figure 5c. From this result it can be concluded that MLP is always producing the same output value, regardless of input vector. For these reasons, this configuration must be omitted. From presented results, it is noticed that MLP evaluation only by using error or bias value can be misleading.

When GA is utilized for design of MLP with four hidden layers ($R_4$), configuration reported in Table 7 is obtained. With these MLP configuration fitness value of 1.4606% is achieved. When B-A analysis is performed for this configuration, it can be noticed that this $R_4$ is performing with bias of 7.4839 and confidence interval in range $d \in [-10.3539, 25.3218]$. It can also be noticed that in the case of lower average values the predicted values are lower than real values, while in the case of higher average values, predicted values are higher than the real values as shown in Figure 5d.

When GA is utilized for design of MLP with five hidden layers ($R_5$), configuration reported in Table 7 is obtained as a best solution. It can be seen that by using this configuration, fitness value of

0.7369% is achieved. When B-A analysis is performed on results achieved with this MLP configuration, it can be seen that it performs with bias of $-1.2609$. It can also be noticed that this approach have a narrower confidence interval ($d \in [-10.7137, 8.1917]$) in comparison to other MLPs that are previously described, as shown in Figure 5e.

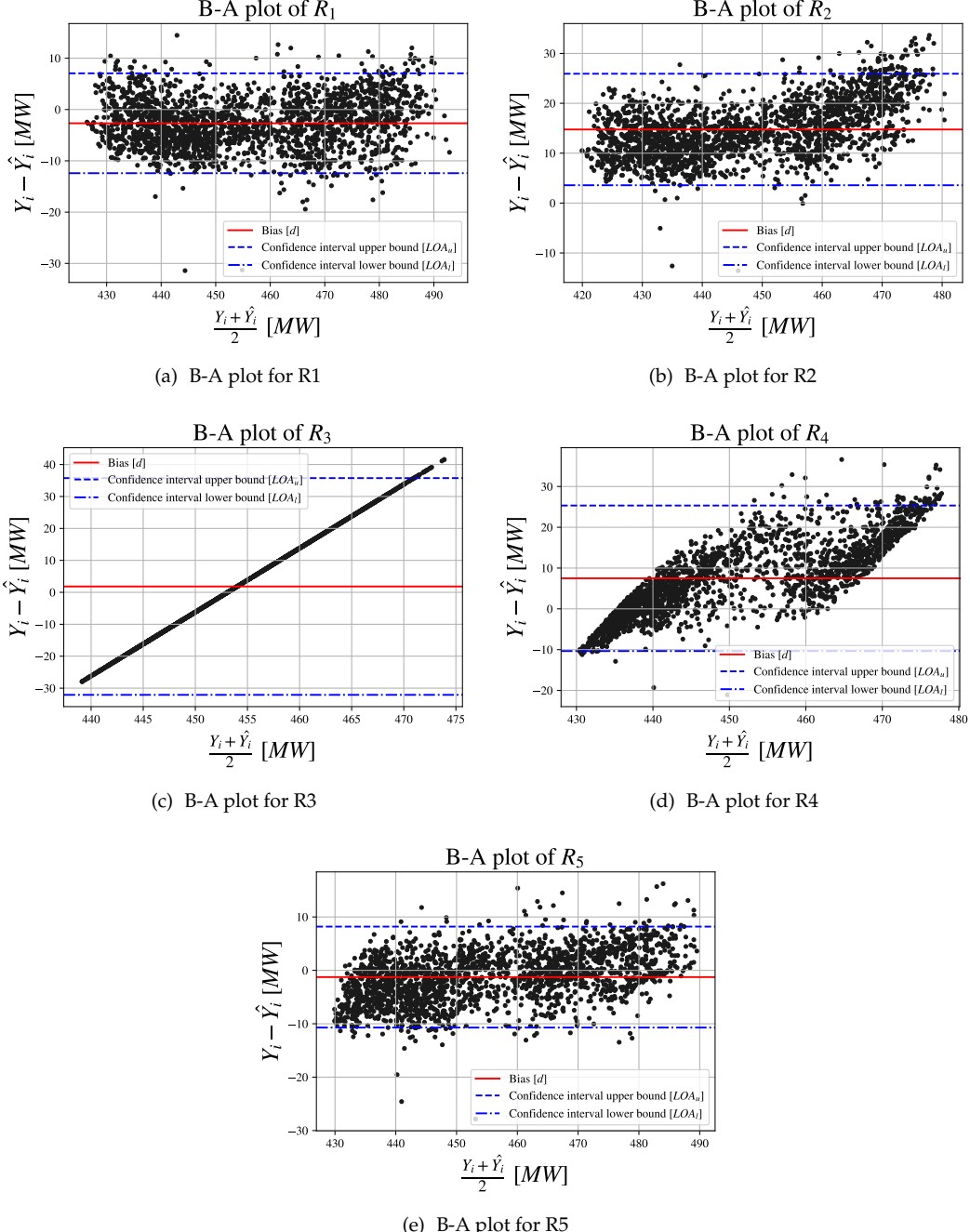

(a) B-A plot for R1

(b) B-A plot for R2

(c) B-A plot for R3

(d) B-A plot for R4

(e) B-A plot for R5

**Figure 5.** Bland-Altman plots for all five MLP configurations designed with GA implementation with $\overline{MRE}$ as a fitness function.

The other approach in GA utilization for the creation of MLP which estimates CCPP electrical power output is to utilize $\overline{MSE}$ as a fitness function. If this approach is used for the creation of MLP with one hidden layer configuration reported in Table 8 is obtained as a best solution. With aforementioned configuration, fitness value of 28.4081 is achieved.

**Table 8.** MLP configurations as a result of the GA implementation with $\overline{MSE}$ as a fitness function.

| | Configuration | | | | |
| --- | --- | --- | --- | --- | --- |
| **Gene** | $S_1$ | $S_2$ | $S_3$ | $S_4$ | $S_5$ |
| Number of epochs | 99 | 55 | 53 | 73 | 55 |
| Number of neurons in the first hidden layer | 35 | 40 | 80 | 30 | 50 |
| Activation function in the first hidden layer | ReLU | Tanh | Sigmoid | ReLU | ReLU |
| Number of neurons in the second hidden layer | - | 100 | 15 | 80 | 15 |
| Activation function in the second hidden layer | - | ReLU | ReLU | Sigmoid | Tanh |
| Number of neurons in the third hidden layer | - | - | 85 | 80 | 100 |
| Activation function in the third hidden layer | - | - | ReLU | Tanh | Tanh |
| Number of neurons in the fourth hidden layer | - | - | - | 80 | 35 |
| Activation function in the fourth hidden layer | - | - | - | ReLU | ReLU |
| Number of neurons in the fifth hidden layer | - | - | - | - | 30 |
| Activation function in the fifth hidden layer | - | - | - | - | ReLU |
| Activation function in the output layer | ReLU | ReLU | ReLU | ReLU | ReLU |
| Batch size | 1278 | 246 | 430 | 820 | 1623 |
| Solver | Adam | AdaGrad | P AdaGrad | RMSprop | Adam |
| Fitness value ($\overline{MSE}$) | 28.4081 | 80.7939 | 39.0868 | 216.6123 | 17.6511 |

When B-A analysis is performed for results achieved with MLP configurations reported in Table 8, it can be seen that MLP with one hidden layer produces estimation of CCPP electrical power output that is biased for $-7.4524$. This bias is a part of confidence interval in range $d \in [-18.7772, 3.8725]$. It can be noticed that lower differences are achieved for samples with higher mean value, as shown in Figure 6a.

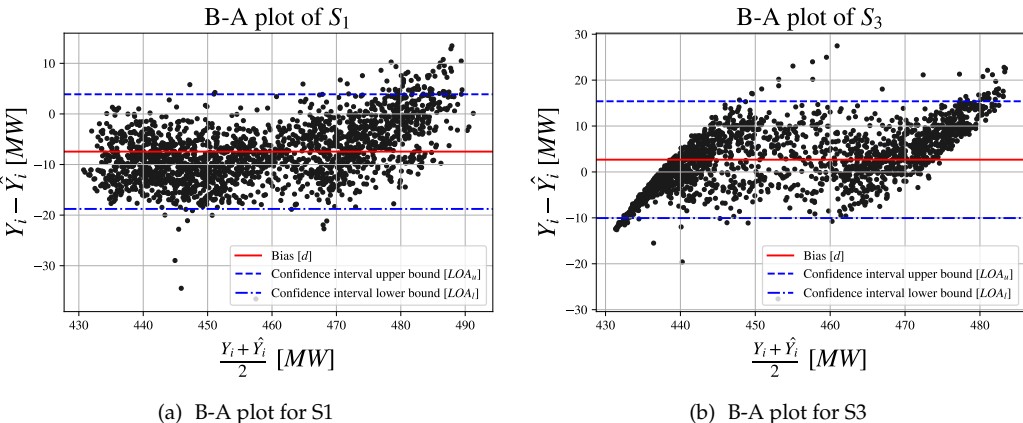

(a) B-A plot for S1　　　　　　　　　　　　　　　　(b) B-A plot for S3

**Figure 6.** Bland-Altman plots for all five MLP configurations designed with GA implementation with $\overline{MSE}$ as a fitness function.

When the results of GA utilization for design of MLP with three hidden layers and $\overline{MSE}$ as a fitness function are observed, it can be noticed that the best fitness value of 39.0868 is achieved with MLP configuration $S_3$ reported in Table 8. When B-A analysis is performed on the results estimated with MLP designed by using parameters presented in Table 8, a biased performance can be noticed. This MLP is performing with bias value of 2.6622 that is a part of the confidence interval $d \in [-10.0570, 15.3814]$. It can also be noticed that results achieved with MLP are lower than real data for lower average values and higher for higher average values, as shown in Figure 6b.

Other configurations presented in Table 8 ($S_2$,$S_4$ and $S_5$) are producing the same output value regardless of input vector, as it is in the case of configuration $R_3$ and its B-A plot shown in Figure 5c. Configuration $S_2$ is performing with bias value of 454.92 with its confidence interval in range

$d \in [421, 488.85]$. Regression performances of configuration $S_4$ are similar to performances of $S_2$. This configuration is performing with bias value of 454.92 and confidence interval in range $d \in [421, 488.85]$. In the case of configuration $S_5$, a bias value of 1.1 is achieved, together with its confidence interval in range $d \in [-35.03, 32.82]$. From presented results, it can be noticed that, in the case of first two configurations, the conclusion in regard of regression performances can be drawn only by observing bias value. That is not a case with configuration $S_5$, where a low bias value can lead to false conclusions. For these reasons, a standard deviation of errors must be taken into account. It can be noticed that all three configurations have high standard deviations of errors ($s_d = 17.3$). For these reasons, aforementioned configurations are omitted and their plots are not displayed.

When all presented results are summarized, it can be noticed that in terms of absolute value, the lowest bias value is achieved in the case of MLP with five hidden layers that is trained by using GA with $\overline{MRE}$ as a fitness function ($R_5$). When MLP with one hidden layer is trained by using the same GA approach ($R_1$), the absolute bias value is noticeably higher. When confidence intervals for aforementioned configurations are compared, it can be seen that, by using configuration $R_5$, absolute value of confidence interval bounds are slightly lower than in case of configuration $R_1$. If $R_5$ is compared to other viable configurations ($R_2$, $R_4$, $S_1$, $S_3$), it can be noticed that in absolute value configuration $R_5$ achieves lower bias value and lover confidence interval bounds, as shown in Figure 7.

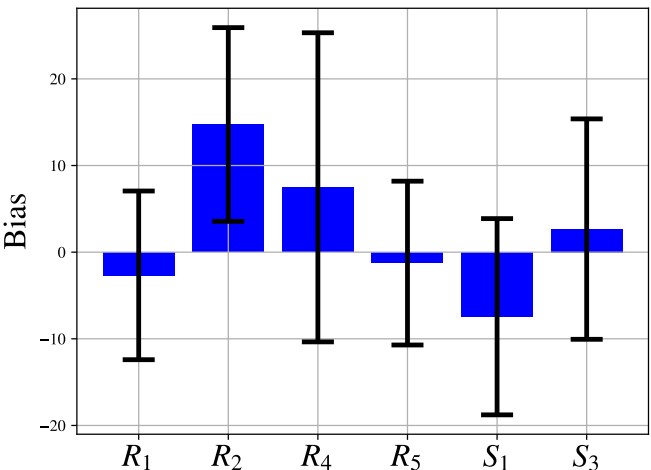

**Figure 7.** Bias and associated confidence interval for each MLP configuration ($R_1$ - MLP with one hidden layer and $\overline{MRE}$ as a fitness function, $R_2$ - MLP with two hidden layers and $\overline{MRE}$ as a fitness function, $R_4$ - MLP with four hidden layers and $\overline{MRE}$ as a fitness function, $R_5$ - MLP with five hidden layers and $\overline{MRE}$ as a fitness function, $S_1$ - MLP with one hidden layer and $\overline{MSE}$ as a fitness function, $S_3$ - MLP with three hidden layers and $\overline{MSE}$ as a fitness function).

As seen from Figure 7 it can be concluded that when using $\overline{MRE}$ as a fitness function there are four of total five MLPs that are viable, while utilization of $\overline{MSE}$ is producing only two viable MLP configurations. Furthermore, it can be noticed that configuration $S_3$ have a significantly higher deviation of error rates in comparison to configurations $R_1$ and $R_5$.

When real $P_e$ data points are compared with $P_e$ estimated by $R_1$ for each 25 samples, it can be noticed that estimation values responds to real value trends, but estimations values often overestimates the real ones especially the data peaks. This comparison is presented in Figure 8a. Comparison of real $P_e$ data points with $P_e$ estimated by $R_5$ shows that overestimation rate is lower (if compared to $R_1$). Furthermore, $R_5$ estimation data better follows real $P_e$ data trends but it also overestimates lower real $P_e$ values, as presented in Figure 8b. Observing estimation performances of $S_3$ follows to conclusion that this MLP underestimates higher and overestimates lower real $P_e$ values, as presented in Figure 8c.

When *RMSE* values achieved with $R_1$, $R_5$ and $S_3$ are compared to *RMSE* values presented in [28], it can be noticed that MLPs developed by GA utilization presented in this paper are achieving significantly lower *RMSE* values, as shown in Table 9.

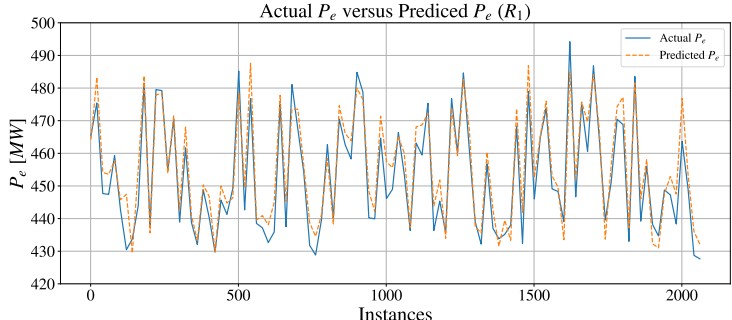

(a) Comparisson between $P_e$ predicted with $R_1$ and real data

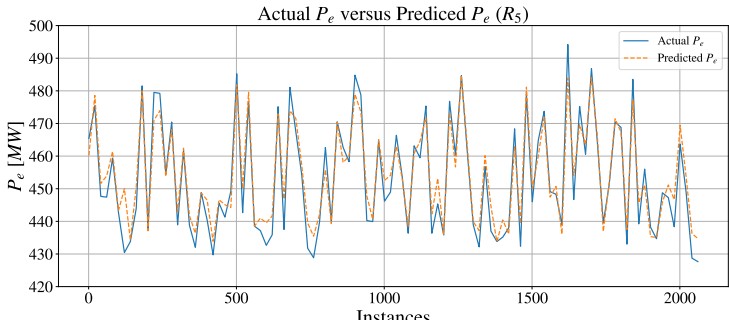

(b) Comparisson between $P_e$ predicted with $R_5$ and real data

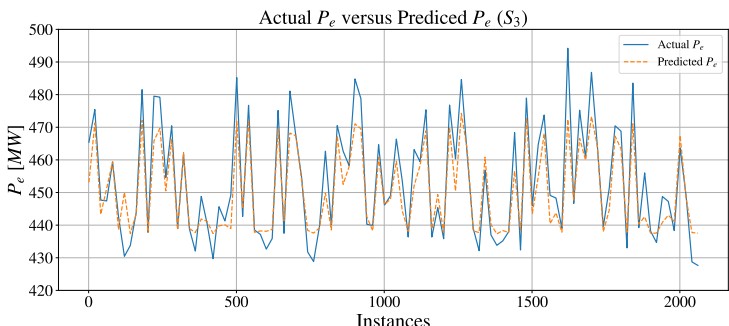

(c) Comparisson between $P_e$ predicted with $S_3$ and real data

**Figure 8.** Comparison between predicted $P_e$ and real data for three MLPs with the best performance: (**a**) $R_1$, (**b**) $R_5$ and (**c**) $S_3$.

**Table 9.** Comparison of results achieved with MLPs designed by utilizing GA and other methods presented in the literature.

| Category | Method | *RMSE* |
|---|---|---|
| Functions | Simple Linear regression | 5.425 |
| | Linear Regression | 4.561 |
| | Least Median Square | 4.968 |
| | Multilayer Perceptron | 5.341 |
| | Radial Basis Funcion Neural Network | 7.501 |
| | Pace Regression | 4.561 |
| | Support Vector Poly Kernel Regression | 4.563 |
| MLPs designed with GA | $R_1$ | 5.07 |
| | $R_5$ | 4.305 |
| | $S_3$ | 4.874 |

Furthermore, it can be concluded that MLPs developed in this paper are achieving lower *RMSE* values in comparison to all methods that are members of Functions category. If 5-fold and 10-fold cross validation procedures are performed, it can be noticed that $R_5$ achieves lowest *RMSE*, regardless of fold numbers. It can also be noticed that, *RMSE* achieved with this configuration have lower standard deviation than other two configurations, as shown in Table 10.

**Table 10.** Results of performed 5-fold and 10-fold cross-validation on configurations $R_1$, $R_5$ and $S_3$.

| | Configuration | Mean | Minimal | Maximal | Standard Deviation |
|---|---|---|---|---|---|
| 5-fold cross-validation | $R_1$ | 5.37 | 5.01 | 5.85 | 0.29 |
| | $R_5$ | 4.31 | 4.16 | 4.43 | 0.09 |
| | $S_3$ | 4.64 | 4.53 | 4.93 | 0.15 |
| 10-fold cross-validation | $R_1$ | 5.27 | 4.66 | 5.99 | 0.45 |
| | $R_5$ | 4.52 | 4.11 | 5.13 | 0.39 |
| | $S_3$ | 4.98 | 4.28 | 5.69 | 0.43 |

When average *RMSE* values presented in Table 10 are compared with *RMSE* values presented in Table 9, it can be noticed that results obtained with 5-fold cross-validation are closely representing results presented in Table 9. That property is particularly emphasized in the case of $R_5$ configuration. Furthermore, it can be noticed that this configuration achieves the lowest standard deviation. In other words, it can be concluded that $R_5$ utilization, alongside lowest *RMSE*, offers the most stable performances in regard of data generalization.

*3.2. Discussion*

When all presented results are summarized, it can be noticed that the best regression performances are achieved if configuration $R_5$ is utilized. This configuration is achieved if GA with $\overline{MRE}$ as a fitness function is implemented for design of MLP with five hidden layers. B-A analysis of aforementioned configuration is showing that this configuration is performing with lower bias and standard deviation of error, in comparison with other configurations. This conclusion can be drawn because this configuration have the narrowest confidence interval of error, as it is shown with B-A plot. Furthermore, when results achieved with $R_3$ are compared with results presented in the literature, it can be noticed that *RMSE* values achieved with this configuration exceeds *RMSE* values achieved in the literature. When presented results are compared to results achieved with $R_1$ and $S_3$, it can be noticed that $R_5$ performs with significantly lower *RMSE* value. If cross-validation technique is utilized in order to evaluate generalization performances of $R_1$, $R_5$ and $S_3$, it can be concluded that configuration $R_5$ is,

again, showing the highest performances. This conclusion could be drawn because of the lowest standard deviation of *RMSE* values. This property is particularly emphasized in the case of 5-fold cross-validation, which leads to the conclusion that this configuration tends towards over-fitting when are trained with larger data sets.

By observing configuration that are omitted during B-A analysis some additional conclusions could be drawn. It can be noticed that configurations $R_3$, $S_2$, $S_4$ and $S_5$ are showing low performances from B-A standpoint while having lower fitness value. This property is particularly emphasized in the case of $S_5$ that achieves low *RMSE* value of 4.2, while B-A analysis shows that these results are not viable from the standpoint of bias and its confidence interval. These results have shown that during GA utilization for MLP design, standard deviation of errors must be taken into account alongside standard measures such as *RMSE*, $\overline{MSE}$ and $\overline{MRE}$.

## 4. Conclusions and Future Work

In this paper, a GA-based method for design of MLPs for CCPP electrical power output estimation is presented. GA is performed by constructing and varying chromosomes that represent MLP parameters. Described method offers a possibility of MLP implementation alongside other regression methods. In comparison to results found in literature, it can be noticed that GA designed MLPs are achieving lower *RMSE* values than other ANN-based methods. GA-based design has produced three MLP architectures that are achieving viable results. From these statements following conclusions could be drawn:

- there is a possibility of GA utilization for design of MLP for CCPP electrical power output,
- presented MLP configurations are performing with lower *RMSE* in comparison to regression methods presented in literature and
- the lowest *RMSE* is achieved if MLP configuration with 5 hidden layers of 80,25,65,75 and 80 nodes, respectively, is utilized. Activation functions used in design of aforementioned MLP are: Logistic Sigmoid in the first layer, Tanh in the second layer and ReLU in all other layers. The best results are achieved if MLP is trained by using Adam solver.

Presented configuration is designed by utilizing GA with $\overline{MRE}$ as a fitness function. It can be noticed that proposed MLP model is one of intermediate complexity, what is in correlation with theoretical knowledge of model selection.

According to presented results, it can be concluded that there is a possibility of heuristic algorithms utilization for design of MLP for CCPP electrical power output estimation.

If results achieved with GA-based MLP are compared to results achieved with other regression methods (algorithms) that are presented in literature, it can be noticed that these methods are still achieving lower *RMSE* values than MLP improved with GA utilization.

Therefore, the future research in observed CCPP electrical power output estimation will be based on utilizing other heuristic algorithms for MLPs design. The final goal will be comparison of the best obtained MLP architectures with different heuristic algorithms, its computational time as well as accuracy and precision of performed predictions - in order to obtain a suitable heuristic algorithm for the investigated problem.

**Author Contributions:** Conceptualization, V.M., Z.C., N.A. and I.L.; Data curation, V.M. and Z.C.; Software, I.L and N.A.; Formal analysis, V.M., Z.C., N.A. and I.L.; Investigation, V.M. and Z.C.; Methodology, V.M., Z.C., N.A. and I.L.; Supervision, V.M., N.A. and Z.C.; Validation, V.M. and Z.C.; Writing–original draft, I.L.; Writing–review and editing, V.M., N.A. and I.L.

**Funding:** This research received no external funding.

**Acknowledgments:** This research has been (partly) supported by the CEEPUS network CIII-HR-0108, European Regional Development Fund under the grant KK.01.1.1.01.0009 (DATACROSS) and University of Rijeka scientific grant uniri-tehnic-18-275-1447.

**Conflicts of Interest:** The authors declare no conflict of interest.

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
