# Peer review of "Genetic Algorithm Approach to Design of Multi-Layer Perceptron for Combined Cycle Power Plant Electrical Power Output Estimation"

_energies, doi:10.3390/en12224352_

Round 1

Reviewer 1 Report

A GA-based method for design of MLPs for CCPP electrical power output estimation is presented.

The paper is well written but contribution is very low.

The paper is too long. It includes very basic textbook content (neural networks and GA). This is supposed to be a research article, so some new contribution is expected.

The original idea is to use GA as a tool in designing the MLP. But it is not clear the motivation. There are so many biologically inspired heuristic optimization algorithms!! (ants, PSO, etc) Authors must give some motivation why GA. In fact, they could use another neural network to obtain the best parameters of the MLS.

With respect the results, they use training and test data, but nothing is said about rigurous cross validation techniques or statistical analysis of the results. It seems a technical report where different arquitectures are tested but no theoretical explanation or seriuos result analysis is carried out,

Author Response

Please find attached responses to Your comments.

Reviewer 2 Report

This paper a genetic algorithm (GA) approach to design of multi-layer perceptron (MLP)for combined cycle power plant power output estimation is presented. The theory is validated on a Combined Cycle Power Plant process. The paper is proper written, but it needs minor improves.

Comments to authors:

- Please add the graph axis with the units of measurement for all the figures.

- The grammar errors should be eliminated in the next version of the paper. Please review the whole paper carefully.

- In section 3 the authors should add more details about how the theory previous sections is applied.

- The state of the art can be improved with references regarding other type of metaheuristic algorithms, perhaps the author could add the following publications:

o Second order intelligent proportional-integral fuzzy control of twin rotor aerodynamic systems, Procedia Computer Science, vol. 139, pp. 372-380, 2018.

o GSP: an automatic programming technique with gravitational search algorithm, Applied Intelligence, vol. 49, no. 4, pp. 1502–1516.

- The authors can add details about their future work.

- A section with the novelty of the current paper should be added in introduction or in the conclusion section.

Author Response

(The authors gave the same response as above.)

Round 2

Reviewer 1 Report

Authors improved the paper.